# Incidence, Risk Factors, and Outcomes of Preterm and Early Term Births: A Population-Based Register Study

**DOI:** 10.3390/ijerph18115865

**Published:** 2021-05-29

**Authors:** Salma Younes, Muthanna Samara, Rana Al-Jurf, Gheyath Nasrallah, Sawsan Al-Obaidly, Husam Salama, Tawa Olukade, Sara Hammuda, Mohamed A. Ismail, Ghassan Abdoh, Palli Valapila Abdulrouf, Thomas Farrell, Mai AlQubaisi, Hilal Al Rifai, Nader Al-Dewik

**Affiliations:** 1Department of Research, Women’s Wellness and Research Center, Hamad Medical Corporation, Doha 3050, Qatar; V-SYounes@hamad.qa (S.Y.); MISMAIL4@hamad.qa (M.A.I.); PABDULROUF@hamad.qa (P.V.A.); TFarrell@hamad.qa (T.F.); 2Department of Psychology, Kingston University London, Kingston upon Thames, London KT1 2EE, UK; M.Samara@kingston.ac.uk (M.S.); sarahammuda@hotmail.com (S.H.); 3Department of Biomedical Science, College of Health Sciences, Member of QU Health, Qatar University, Doha 2713, Qatar; 199660690@student.qu.edu.qa (R.A.-J.); gheyath.nasrallah@qu.edu.qa (G.N.); 4Obstetrics and Gynecology Department, Women’s Wellness and Research Center, Hamad Medical Corporation, Doha 3050, Qatar; Salobaidly@hamad.qa; 5Department of Pediatrics and Neonatology, Neonatal Intensive Care Unit, Newborn Screening Unit, Women’s Wellness and Research Center, Hamad Medical Corporation, Doha 3050, Qatar; hsalama1@hamad.qa (H.S.); TOlukade@hamad.qa (T.O.); Gabdoh@hamad.qa (G.A.); MALQUBAISI@hamad.qa (M.A.); Halrifai@hamad.qa (H.A.R.); 6Department of Pharmacy, Women’s Wellness and Research Center, Hamad Medical Corporation, Doha 3050, Qatar; 7Interim Translational Research Institute (iTRI), Hamad Medical Corporation (HMC), Doha 3050, Qatar; 8Faculty of Health and Social Care Sciences, Kingston University, St. George’s University of London, London KT1 2EE, UK; 9Clinical and Metabolic Genetics, Department of Pediatrics, Hamad General Hospital, Hamad Medical Corporation, Doha 3050, Qatar; 10College of Health and Life Science (CHLS), Hamad Bin Khalifa University (HBKU), Doha 34110, Qatar

**Keywords:** preterm birth (PTB), prematurity, late preterm birth (Late PTB), early term birth (ETB), incidence, risk factors, outcomes, Qatar

## Abstract

Preterm birth (PTB) and early term birth (ETB) are associated with high risks of perinatal mortality and morbidity. While extreme to very PTBs have been extensively studied, studies on infants born at later stages of pregnancy, particularly late PTBs and ETBs, are lacking. In this study, we aimed to assess the incidence, risk factors, and feto-maternal outcomes of PTB and ETB births in Qatar. We examined 15,865 singleton live births using 12-month retrospective registry data from the PEARL-Peristat Study. PTB and ETB incidence rates were 8.8% and 33.7%, respectively. PTB and ETB in-hospital mortality rates were 16.9% and 0.2%, respectively. Advanced maternal age, pre-gestational diabetes mellitus (PGDM), assisted pregnancies, and preterm history independently predicted both PTB and ETB, whereas chromosomal and congenital abnormalities were found to be independent predictors of PTB but not ETB. All groups of PTB and ETB were significantly associated with low birth weight (LBW), large for gestational age (LGA) births, caesarean delivery, and neonatal intensive care unit (NICU)/or death of neonate in labor room (LR)/operation theatre (OT). On the other hand, all or some groups of PTB were significantly associated with small for gestational age (SGA) births, Apgar < 7 at 1 and 5 min and in-hospital mortality. The findings of this study may serve as a basis for taking better clinical decisions with accurate assessment of risk factors, complications, and predictions of PTB and ETB.

## 1. Introduction

Preterm birth (PTB) is considered the main cause of perinatal mortality and morbidity in industrialized countries [1]. Many PTB-born children who survive face adverse short-term and long-term consequences, including behavioral. health related quality of life, cognitive, eating and neurological impairments, and chronic conditions, which in some cases, can lead to death [2,3,4,5,6,7,8].

In the past, a pregnancy that lasted anywhere between 37 and 42 weeks was referred to as a term pregnancy [9,10]. However, driven by the growing body of evidence that significant differences exist in the outcomes of infants delivered within this 5-week interval, the definitions for term pregnancies have been updated [9,11]. In 2013, a full-term pregnancy (FTB) has been redefined as a pregnancy that lasts between 39 weeks, 0 days and 40 weeks 6 days, whereas the period between 37 weeks to 38 weeks, 6 days of gestation is now referred to as early term birth (ETB) [9,11]. PTBs are not the only gestational age (GA) subgroup at risk of adverse clinical outcomes [12,13]. Recent evidence suggests that ETBs are associated with high risk of mortality, more admissions to neonatal intensive care units [13], as well as conferring a high risk for respiratory diseases and neurological disabilities [14].

Over the past 20 years, the rate of PTB has been escalating steadily and alarmingly. Of 65 countries around the world, all but three countries have shown an increase in PTB rates over the past two decades [1]. The estimated global PTB rate in 2014 was 10.6%, corresponding to approximately ~14.8 million live preterm births, of which ~12 million (81.1%) occurred in Asia and sub-Saharan Africa [15]. Over 75% of all perinatal mortality and 50% of perinatal and long-term morbidities are associated with PTBs [16,17]. ETBs are more common than PTBs, with international prevalence rates ranging from 15% to 30% [18,19]. In developed countries, the pooled rate of ETBs in in the period 2006–2014 was estimated to range from 16.8% in Finland to 26.9% in the USA [18]. However, the proportion of ETBs in many developing countries remains unclear. Thus, there is a lack of evidence concerning the impact of ETB on maternal and neonatal outcomes.

PTBs and ETBs are believed to involve both genetic and environmental factors. Causal factors linked to PTB include medical conditions [20], genetic influences [21,22,23], environmental exposure [24,25], infertility treatments [26,27], as well as behavioral and socioeconomic factors [28,29,30]. While extreme to very PTBs have been extensively studied, and the associated morbidities are well-documented [31,32,33], studies on infants born at later stages of pregnancy, particularly late PTBs and ETBs, are lacking. Thus, in this population-based study, we aimed to assess the incidence of PTB and ETB over a period of 12 months in Qatar, and examine the associated neonatal and maternal risk factors, as well as the feto-maternal consequences in the perinatal-neonatal period.

## 2. Materials and Methods

### 2.1. Study Design and Data Collection

This study is a 12-month retrospective population-based study conducted using registry data from the PEARL-Peristat Study, Qatar. This population-based registry was designed using routinely collected hospital data for parturient women and their offspring. The study was approved by the Hamad Medical Corporation (HMC) Institutional Review Board (IRB), with a waiver of consent.

We included singleton live births at 24^+0^ weeks gestation and above, whose mothers delivered between April 2017 and March 2018 at the Women’s Wellness and Research Centre (WWRC) in HMC, the main national hospital and provider of secondary and tertiary health care facilities in Qatar. Stillbirths were excluded, as there was no certainty of the presence or absence of PTBs. Out of 15,872 identified singleton births, gestational age was missing for 7 infants and thus these were excluded. Finally, a total of 15,865 singleton births were examined.

### 2.2. Neonatal Factors

Gestational age was classified into preterm and term births; preterm births were further categorized into extremely to very PTB (<32 weeks), moderate PTB (32 to <34 weeks), and late PTB (34 to <37 weeks). Moreover, in accordance with established international definitions [12], we further discriminated between ETB (37 to <39) and FTB (39 to <42).

Baby gender was categorized into male, female, and ambiguous. Birth weight was categorized as low and normal (≤2499 g and ≥2500 g, respectively). In addition, appropriate for gestational age (AGA), small for gestational age (SGA), and large for gestational age (LGA) were defined as those with estimated fetal weight between 10th percentile and 90th percentile, below the 10th percentile, or above the 90th percentile, respectively. Low Apgar scores at 1 min and at 5 min were defined as any score lower than 7 [34]. Baby outcome was categorized into discharged alive or in-hospital mortality, while baby disposition was categorized into postnatal ward and Neonatal Intensive Care Unit (NICU) or death in Labor Room/ Operation Theatre (LR/OT).

### 2.3. Maternal Factors

Maternal age was categorised into young (<20 years), normal (20–34 years), advanced (35–39 years), very advanced (40–44 years), and extremely advanced (≥45 years) age ranges. Parity was classified into nulliparous or parity ≥ 1. Nationality was grouped into Qatari (*n*: 4941, 32%), other Arabs based on the UNESCO list of Arab countries (*n*: 6306, 40%), and other nationalities (Asian: *n*: 3711, 23%; Europeans and North and South Americans: *n*: 151, 1%; others: *n*: 752, 4%). Consanguinity was coded as yes (the mother and the father are related to each other in any level of relatedness) or no. Educational level was grouped into three categories including elementary and below, secondary school or high school, and college/university or above. Employment status was categorised into employed or unemployed. History of PTB and history of smoking were coded as yes versus no. To reduce selection bias from PTB history variable, we excluded women who were marked as having a history of multiple birth from this variable. The risk of PTB is higher with multiple pregnancies.

Women were categorized according to their glycemic status into diabetic and non-diabetic, and further categorized into pregestational diabetics (PGDM), gestational diabetics (GDM), and non-diabetics (no data on Type 1 or 2 were recorded). Chronic hypertension was coded as yes or no. Women were also categorized into obese and non-obese using pre-pregnancy and the first 12 weeks of pregnancy (pre- and early-pregnancy; PP obesity) BMI. This was determined by calculating the BMI during that period, calculated from height and weight. The variable was then categorized into non-obese (<30 kg/m^2^) and obese (BMI ≥ 30 kg/m^2^) following NHLBI/WHO guidelines [35,36].

Pregnancy mode was defined as spontaneous or assisted (including ovulation induction, invitro fertilisation, intracytoplasmic sperm injection, intra uterine insemination, and others). Delivery mode was categorized into vaginal and caesarean.

### 2.4. Statistical Analysis

Statistical analysis using IBM SPSS 26 software (SPSS, Chicago, IL, USA) was conducted. All categorical and binary variables were reported as numbers and percentages. The overall incidence of PTB and ETB, risk factors, and outcomes were analyzed using Chi-Square analysis on the differences between each PTB and ETB groups in one hand and FTB on the other hand. Regression analyses were performed in two stages. Risk factors associated with PTB and ETB were examined and the associations with the outcomes were quantified; they were reported as odds ratios (OR) and 95% confidence intervals (CI).

In stage one, we used univariate logistic regression to investigate the association of maternal and neonatal risk factors with PTBs and ETBs in comparison to FTB. Multiple logistic regression was then performed including the significant variables with *p* < 0.05 from the univariate analysis. PP obesity was excluded from the multiple logistic regression due to missing values. In stage two, univariate logistic regression was performed to investigate the association of PTBs and ETBs in comparison with FTB as predictors of several neonatal and maternal outcomes. After that multiple logistic regression was performed adjusting for the variables that were significant from the univariate analysis at the first stage. The independent risk factors included in the final model were reported as ORs and 95% CIs. Statistical significance was defined as *p* < 0.05.

## 3. Results

### 3.1. Characteristics of the Study Population and Differences between Pre and Early Term Groups in Comparison to Full Term Group

A total of 15,872 singleton live births registered in the PEARL database from April 2017 to March 2018 were identified, of which 15,865 singleton live births were examined (Figure 1, Appendix A).

The maternal characteristics and newborn distribution of the overall study population are shown in Table 1. Table 1 shows the differences between each PTB and ETB group in comparison to FTB in relation to each demographic and medical variable. PTB and ETB incidence rates were 8.8% and 33.7%, respectively. In-hospital mortality was 16.9% among PTB infants and 0.2% among ETB infants (Table 1, Appendix A).

### 3.2. Risk Factors Associated with PTB and ETB

The regression analysis shows that for extreme to very PTB, advanced maternal age (aOR, 1.81; 95% CI, 1.15–2.83), very advanced maternal age (aOR, 2.39; 95% CI, 1.12–5.11), parity ≥ 1 (aOR, 0.34; 95% CI, 0.23–0.51), PGDM (aOR, 7.37; 95% CI, 2.97–18.31), chromosomal/congenital abnormalities (aOR, 15.52; 95% CI, 8.91–27.02), history of preterm birth (aOR, 7.23; 95% CI, 4.44–11.77), and assisted pregnancy (aOR, 2.49; 95% CI, 1.19–5.21) were found to be independent predictors in the univariate and the multivariate analyses (Table 2, Appendix A). However, ethnic group, chronic hypertension, baby gender, and smoking history status became non-significant in the adjusted model (Table 2, Appendix A).

For moderate PTB, young maternal age (aOR, 2.76; 95% CI, 1.26–6.07), other Arab nationalities (aOR, 0.6; 95% CI, 0.4–0.89), GDM (aOR, 1.47; 95% CI, 1.03–2.1), PGDM (aOR, 5.03; 95% CI, 2.08–12.17), chronic hypertension (aOR, 6.5; 95% CI, 2.92–14.46), chromosomal/congenital abnormalities (aOR, 7.9; 95% CI, 4.14–15.04), assisted pregnancy (2.82; 95% CI, 1.38–5.79), preterm history (aOR, 4.7; 95% CI, 3.04–7.28), and male baby gender (aOR, 1.46; 95% CI, 1.06–2.03) were found to be significant predictors in the univariate and the multivariate analyses (Table 2, Appendix A). However, mothers with advanced (35–39 years) and very advanced (40–44 years) ages became non-significant after adjustment (Table 2, Appendix A).

For late PTB, advanced maternal age (aOR, 1.39; 95% CI, 1.05–1.86), other Arab nationalities (aOR, 0.64; 95% CI, 0.49–0.83), level of education of elementary and below (aOR, 1.46; 95% CI, 1.02–2.08), GDM (aOR, 1.61; 95% CI, 1.27–2.05), PGDM (aOR, 6.55; 95% CI, 3.17–13.52), chromosomal/congenital abnormalities (aOR, 3.9; 95% CI, 1.97–7.70), preterm history (aOR, 3.5; 95% CI, 2.48–4.94), and assisted delivery (aOR, 3.11; 95% CI, 1.90–5.08) were found to be independent predictors in the univariate and the multivariate analyses (Table 2, Appendix A). However, advanced maternal age (35–39 years), extremely advanced maternal age (≥45 years), chronic hypertension, and male gender became non-significant in the adjusted model (Table 2, Appendix A).

For ETB, advanced maternal age (aOR, 1.18; 95% CI, 1.07–1.31), very advanced maternal age (aOR, 1.53; 95% CI, 1.28–1.82), extremely advanced maternal age (aOR, 2.36; 95% CI, 1.18–4.71), parity ≥ 1 (aOR, 1.21; 95% CI, 1.11–1.32), other Arab nationalities (aOR, 0.89; 95% CI, 0.81–0.96), GDM (aOR, 1.8; 95% CI, 1.67–1.94), PGDM (aOR, 9.72; 95% CI, 7.07–13.35), chronic hypertension (aOR, 2.39; 95% CI, 1.67–3.43), male babies (1.2; 95% CI, 1.11–1.28), preterm history (aOR, 1.95; 95% CI, 1.68–2.26), and assisted pregnancies (aOR, 1.61; 95% CI, 1.28–2.02) were found to be independent predictors in the univariate and the multivariate analyses (Table 2, Appendix A). However, chromosomal/congenital abnormalities became non-significant in the adjusted model (Table 2, Appendix A).

### 3.3. Adverse Outcomes Associated with PTB and ETB

Extreme to very PTB significantly predicted all assessed outcomes; LBW (aOR, 5669.36; 95% CI, 1317.88–24388.98), SGA (aOR, 4.11; 95% CI, 2.55–6.64), LGA (aOR, 3.8; 95% CI, 2.5–5.79), caesarean delivery (aOR, 5.26; 95% CI, 3.71–7.47), low Apgar < 7 at 1 min (aOR, 52.3; 95% CI, 33.89–80.71) and at 5 min (aOR, 69.2; 95% CI, 20.54–233.09), in-hospital mortality (aOR, 84.73; 95% CI, 28.87–248.69), and NICU/death in LR/OT (aOR, 2417.24; 95% CI, 332.79–17557.67) in the univariate and the adjusted models (Table 3, Appendix A).

Moderate PTB significantly predicted all assessed outcomes; LBW (aOR, 1021.64; 95% CI, 520.79–2004.19), SGA (aOR, 3.78; 95% CI, 2.38–6.00), LGA (aOR, 2.44; 95% CI, 1.63–3.64), caesarean delivery (aOR, 4.54; 95% CI, 3.26–6.34), low Apgar < 7 at 1 min (aOR, 7.73; 95% CI, 4.43–13.50) and at 5 min (aOR, 9.81; 95% CI, 2.01–47.91), in-hospital mortality (aOR, 27.76; 95% CI, 7.52–102.48), and NICU/death in LR/OT (aOR, 274.38; 95% CI, 132.59–567.78) in the univariate and the adjusted models (Table 3, Appendix A).

Late PTB significantly predicted LBW (aOR, 57.41; 95% CI, 40.1–82.19), SGA (aOR, 1.9; 95% CI, 1.28–2.82), LGA (aOR, 1.87; 95% CI, 1.39–2.5), caesarean delivery (aOR, 2.47; 95% CI, 1.96–3.12), and NICU/death in LR/OT (aOR, 8.48; 95% CI, 6.37–11.28) in the univariate and the adjusted models. However, low Apgar < 7 at 1 min and at 5 min, and in-hospital mortality became non-significant after adjustment (Table 3, Appendix A).

ETB significantly predicted LBW (aOR, 5.96; 95% CI, 4.83–7.35), LGA (1.37; 95% CI, 1.24–1.52), caesarean delivery (aOR, 2.14; 95% CI, 1.98–2.32), Apgar < 7 at 1 min (aOR, 0.71; 95% CI, 0.51–0.99), and NICU/death in LR/OT (aOR, 1.31; 95% CI, 1.15–1.51) in the univariate and the adjusted models. However, low Apgar < 7 at 5 min, and in-hospital mortality became non-significant after adjustment (Table 3, Appendix A).

## 4. Discussion

Our population-based study showed a PTB incidence of 88 per 1000 total singleton births (8.8%), and ETB incidence was estimated to be 337 per 1000 total singleton births (33.7%), between April 2017 and March 2018. Several factors were shared among the different gestational age groups, including advanced maternal age, PGDM, assisted delivery, and preterm history, which conferred a relatively higher risk for PTB and ETB in comparison to FTB (Table 2, Appendix A). In-hospital mortality was highest within the extreme to very PTB (11.2%) group in comparison to the other PTB groups (moderate: 4.4%; late: 1.3%). Not only PTB, but also ETB was significantly associated with higher rates of caesarean deliveries, LBW, LGA, and admission to NICU/or death in LR/OT.

The estimated incidence for PTB (8.8%) is higher than that reported in Qatar back in 2014 (4.48%, UI: 3.02–6.42) [16], comparable to that reported in Europe, and Latin America and the Caribbean (8.8% and 9.8%, respectively) [11], yet, it is lower than that reported in North America (11.2%), Northern Africa (13.4%), Oceania (10.0%), and Sub-Saharan Africa (12.0%) [11]. In regard to ETB, there is a dearth of epidemiological studies assessing the incidence, risk factors, and outcomes to be able to establish valid comparisons [37].

In this study, advanced maternal age, PGDM, assisted delivery, and preterm history were independent predictors of both PTB and ETB (Table 2, Appendix A). These are all well-established risk factors for PTB and ETBs among different racial and ethnic groups [21,22,38,39]. Mothers with chromosomal or congenital abnormalities were more likely to deliver PTB but not ETB, after adjusting for the confounding factors. This indicates that the trend of the risk distribution for PTB delivery in mothers with chromosomal or congenital abnormalities may differ according to the distribution of gestational age. 

In this study, both PTB and ETB significantly predicted low birth weight, large-for-gestational-age (LGA) births, caesarean delivery, admission to neonatal intensive care unit (NICU), and deaths of neonate in labor room (LR)/or operation theatre (OT) (Table 3). Globally, PTB complications are the leading cause of mortality among children aged < 5 years, responsible for an estimated 1 million deaths in 2015 [40]. There are huge geographical variations in the rates of PTB mortality and absolute number of deaths due to complications related to PTBs [41]. As gestational age decreases, mortality rates increase, and infants who are both PTB and SGA are at even higher risk [42,43]. In high-income countries, advancements in healthcare have helped improve survival and long-term adverse consequences in very and extremely PTB-born children [44]. In developed countries, extremely PTB-born infants are estimated to have a 90% survival chance, nevertheless, they may suffer long-term neurological, and physical disabilities. On the other hand, only 10% of extremely PTB-born infants are estimated to survive in low-income countries [45].

Despite the consequences and burden caused by these outcomes, most of them can be prevented [35]. It is estimated that three-quarters of the global deaths due to PTBs could be prevented with current, cost-effective interventions [40]. According to reports from high-income countries, it is estimated that most prematurity cases can be offered care that could be lifesaving or that would reduce adverse events. It is noteworthy to mention that cesarean sections in cases of PTB labor can be protective, but can also lead to significant morbidities among both the mothers and their babies, and thus, the ideal delivery mode for PTB singletons remains controversial [46].

This population-based study was performed using data retrieved from the PEARL-Peristat Study (Perinatal Neonatal Registry), based on predesigned hospital data pertaining to mothers and newborns [47]. This database is large enough that the sample size is representative of births in Qatar [47]. Additionally, since the concept of ETB is a relatively new one, this study is one of only a few studies to examine the risk factors and outcomes associated with ETB.

## 5. Conclusions

This large population-based study is the first to assess the incidence, risk factors, and outcomes associated with PTB and ETB in Qatar and one of the very few studies to assess the risk factors and outcomes associated with ETB worldwide. The findings of this study may serve as a basis to help make better clinical decisions with accurate assessment of risk factors, complications, and realistic predictions related to PTB and ETB, which should ultimately provide a way forward for precision health and to help reduce the burden and the consequences associated with PTB and ETB.

## Figures and Tables

**Figure 1 ijerph-18-05865-f001:**
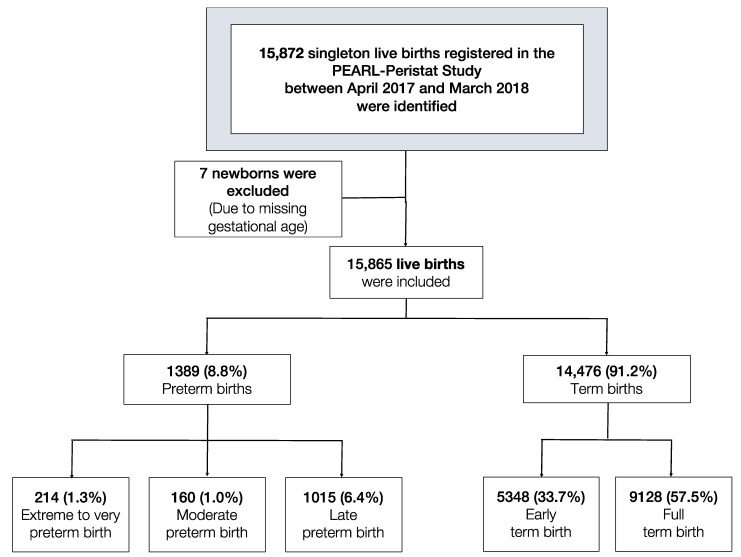
Flow chart of the study live births: term and preterm numbers and percentages.

**Table 1 ijerph-18-05865-t001:** Characteristics of the study population according to gestational age groups: differences between each PTB and ETB group in comparison to FTB in relation to various demographic and medical variables.

	Extreme to Very PTB (*n* = 214)		Moderate PTB (*n* = 160)		Late PTB (*n* = 1015)		ETB (*n* = 5348)		FTB (*n* = 9128) (Ref.)
	*n* (%)	*p* Value	*n* (%)	*p* Value	*n* (%)	*p* Value	*n* (%)	*p* Value	*n* (%)
Maternal age		**<0.001**		**<0.001**		**<0.001**		**<0.001**	
Normal (20–34 yr)	149 (69.6)		107 (66.9)		718 (70.7)		3973 (74.3)		7402 (81.1)
Young (<20 yr)	5 (2.3)		7 (4.4)		20 (2)		99 (1.9)		219 (2.4)
Advanced (35–39 yr)	46 (21.5)		36 (22.5)		217 (21.4)		937 (17.5)		1209 (13.2)
Very advanced (40–44 yr)	14 (6.5)		9 (5.6)		53 (5.2)		310 (5.8)		285 (3.1)
Extremely advanced (≥45 yr)	0 (0)		1 (0.6)		7 (0.7)		29 (0.5)		13 (0.1)
Parity		**0.001**		0.283		0.105		**<0.001**	
Nulliparous	85 (39.7)		53 (33.1)		272 (26.8)		1232 (23)		2668 (29.2)
Parity ≥ 1	129 (60.3)		107 (66.9)		743 (73.2)		4116 (77)		6460 (70.8)
Nationality		**<0.05**		**<0.05**		**<0.001**		**0.001**	
Qatari	61 (28.5)		57 (35.6)		343 (33.8)		1730 (32.4)		2750 (30.1)
Other Arabs	75 (35)		48 (30)		349 (34.4)		2048 (38.3)		3786 (41.5)
Other Nationalities	78 (36.4)		55 (34.4)		323 (31.8)		1569 (29.3)		2589 (28.4)
Consanguineous		0.161		0.651		0.163		0.634	
No	42 (75)		27 (62.8)		217 (62.4)		1435 (66.7)		2444 (66.1)
Yes	14 (25)		16 (37.2)		131 (37.6)		717 (33.3)		1255 (33.9)
Education		0.570		0.168		**<0.05**		0.248	
Elementary and below	4 (6.5)		1 (2.2)		47 (12.4)		205 (9)		355 (8.9)
Secondary/Highschool	23 (37.1)		12 (26.7)		130 (34.3)		673 (29.5)		1254 (31.5)
University or above	35 (56.5)		32 (71.1)		202 (53.3)		1404 (61.5)		2373 (59.6)
Diabetes status		**<0.001**		**<0.001**		**<0.001**		**<0.001**	
No DM	161 (75.2)		103 (64.4)		613 (60.4)		3259 (60.9)		6981 (76.5)
GDM	43 (20.1)		48 (30)		348 (34.3)		1829 (34.2)		2099 (23)
PGDM	10 (4.7)		9 (5.6)		54 (5.3)		260 (4.9)		48 (0.5)
Chronic hypertension		**<0.001**		**<0.001**		**<0.001**		**<0.001**	
No	205 (95.8)		149 (93.1)		982 (96.7)		5243 (98)		9078 (99.5)
Yes	9 (4.2)		11 (6.9)		33 (3.3)		105 (2)		50 (0.5)
PP obesity		**<0.05**		0.529		**<0.001**		**<0.001**	
No	43 (58.9)		33 (67.3)		262 (62.7)		1403 (63.4)		2369 (71.4)
Yes	30 (41.1)		16 (32.7)		156 (37.3)		811 (36.6)		947 (28.6)
Baby gender		**<0.01**		**<0.05**		**<0.001**		**<0.001**	
Male	125 (58.4)		95 (59.4)		557 (54.9)		2843 (53.2)		4482 (49.1)
Female	89 (41.6)		65 (40.6)		457 (45)		2503 (46.8)		4646 (50.9)
Ambiguous	0 (0)		0 (0)		1 (0.1)		1 (0)		0 (0)
Chromosomal/congenital abnormality		**<0.001**		**<0.001**		**<0.001**		**<0.05**	
No	185 (86.4)		148 (92.5)		968 (95.4)		5271 (98.6)		9032 (98.9)
Yes	29 (13.6)		12 (7.5)		47 (4.6)		77 (1.4)		96 (1.1)
Smoking		**<0.05**		0.308		0.452		0.862	
No	148 (97.4)		120 (98.4)		745 (98.9)		4117 (99.2)		6932 (99.2)
Yes	4 (2.6)		2 (1.6)		8 (1.1)		32 (0.8)		56 (0.8)
Preterm history		**<0.001**		**<0.001**		**<0.001**		**<0.001**	
No	172 (80.4)		130 (81.3)		860 (84.7)		4895 (91.5)		8758 (95.9)
Yes	42 (19.6)		30 (18.8)		155 (15.3)		453 (8.5)		370 (4.1)
Employment status		0.477		0.346		0.357		0.151	
Employed	54 (100)		42 (97.7)		344 (98.6)		2159 (98.7)		3737 (99.1)
Unemployed	0 (0)		1 (2.3)		5 (1.4)		29 (1.3)		35 (0.9)
Birth weight		**<0.001**		**<0.001**		**<0.001**		**<0.001**	
≤2499 g	210 (98.6)		148 (92.5)		432 (42.6)		375 (7)		127 (1.4)
≥2500 g	3 (1.4)		12 (7.5)		583 (57.4)		4972 (93)		8996 (98.6)
Fetal weight by GA		**<0.001**		**<0.001**		**<0.001**		**<0.001**	
AGA	120 (56.3)		96 (60)		682 (67.2)		4126 (77.2)		6453 (81.6)
SGA	41 (19.2)		28 (17.5)		112 (11)		250 (4.7)		451 (5.7)
LGA	52 (24.4)		36 (22.5)		221 (21.8)		971 (18.2)		1002 (12.7)
Pregnancy mode		**<0.001**		**<0.01**		**<0.001**		**<0.001**	
Spontaneous	202 (94.4)		150 (94.3)		961 (95.1)		5159 (97)		8906 (98.1)
Assisted	12 (5.6)		9 (5.7)		49 (4.9)		157 (3)		169 (1.9)
Delivery mode		**<0.001**		**<0.001**		**<0.001**		**<0.001**	
Vaginal	77 (36)		63 (39.4)		502 (49.5)		3236 (60.5)		7169 (78.5)
Caesarean	137 (64)		97 (60.6)		513 (50.5)		2112 (39.5)		1959 (21.5)
Apgar < 7 at 1 min		**<0.001**		**<0.001**		**<0.001**		**<0.05**	
No	112 (53.1)		137 (87.3)		956 (94.5)		5280 (99)		8975 (98.6)
Yes	99 (46.9)		20 (12.7)		56 (5.5)		56 (1)		131 (1.4)
Apgar < 7 at 5 min		**<0.001**		**<0.001**		**<0.001**		0.251	
No	196 (92.9)		154 (98.1)		1003 (99)		5328 (99.8)		9102 (99.9)
Yes	15 (7.1)		3 (1.9)		10 (1)		9 (0.2)		9 (0.1)
Baby outcome		**<0.001**		**<0.001**		**<0.001**		0.052	
Discharged alive	190 (88.8)		153 (95.6)		1002 (98.7)		5335 (99.8)		9118 (99.9)
In-hospital mortality	24 (11.2)		7 (4.4)		13 (1.3)		13 (0.2)		10 (0.1)
Baby disposition		**<0.001**		**<0.001**		**<0.001**		**<0.001**	
Postnatal ward	1 (0.5)		8 (5)		640 (63.1)		4853 (90.8)		8528 (93.4)
NICU or death in LR/OT	213 (99.5)		152 (95)		375 (36.9)		494 (9.2)		600 (6.6)

Abbreviations: Ref, reference group; PTB, preterm birth; ETB, early term birth; DM, diabetes mellitus; GDM, gestational diabetes mellitus; PGDM, pre-gestational diabetes mellitus; PP, pre- and early-pregnancy; AGA, appropriate for gestational age; SGA, small for gestational age; LGA, large for gestational age; Apgar, appearance, pulse, grimace, activity, and respiration; NICU, neonatal intensive care unit; LR, labour room; OT, operation theatre. **Bold** values denote statistically significant differences between each PTB and ETB group in one hand and FTB on the other hand.

**Table 2 ijerph-18-05865-t002:** Multivariate regression analyses of the risk factors associated with PTB and ETB.

Risk Factors	Extreme to Very PTB (*n* = 214)	Moderate PTB (*n* = 160)	Late PTB (*n* = 1015)	ETB (*n* = 5348)
	aOR ^a^ (95% CI)	aOR ^b^ (95% CI)	aOR ^c^ (95% CI)	aOR ^d^ (95% CI)
Maternal age				
Normal (20–34 yr)	ref	ref	ref	ref
Young (<20 yr)	0.96 (0.34–2.72)	**2.76 (1.26–6.07) ‡**	1.01 (0.48–2.13)	1.02 (0.8–1.31)
Advanced (35–39 yr)	**1.81 (1.15–2.83) ‡**	1.48 (0.98–2.23)	**1.39 (1.05–1.86) ‡**	**1.18 (1.07–1.31) ***
Very advanced (40–44 yr)	**2.39 (1.12–5.11) ‡**	1.77 (0.87–3.63)	1.57 (0.91–2.71)	**1.53 (1.28–1.82) ***
Extremely advanced (≥45 yr)	NA	1.64 (0.16–16.9)	4.97 (0.79–31.38)	**2.36 (1.18–4.71) ‡**
Parity				
Nulliparous	ref			ref
Parity ≥ 1	**0.34 (0.23–0.51) ***			**1.21 (1.11–1.32) ***
Nationality				
Qataris	0.78 (0.51–1.2)	0.87 (0.59–1.28)	0.8 (0.61–1.05)	0.99 (0.9–1.08)
Other Arabs	0.73 (0.48–1.09)	**0.6 (0.4–0.89) ‡**	**0.64 (0.49–0.83) ***	**0.89 (0.81–0.96) †**
Other Nationalities	ref	ref	ref	ref
Education				
Elementary and below			**1.46 (1.02–2.08) ‡**	
Secondary/Highschool			1.25 (0.98–1.59)	
University or above			ref	
Diabetes status				
No DM	ref	ref	ref	ref
GDM	0.88 (0.58–1.33)	**1.47 (1.03–2.1) ‡**	**1.61 (1.27–2.05) ***	**1.8 (1.67–1.94) ***
PGDM	**7.37 (2.97–18.31) ***	**5.03 (2.08–12.17) ***	**6.55 (3.17–13.52) ***	**9.72 (7.07–13.35) ***
Chronic hypertension				
No	ref	ref	ref	ref
Yes	2.97 (0.96–9.18)	**6.5 (2.92–14.46) ***	1.14 (0.37–3.49)	**2.39 (1.67–3.43) ***
Baby gender				
Female	ref	ref	ref	ref
Male	1.39 (0.99–1.95)	**1.46 (1.06–2.03) ‡**	1.12 (0.9–1.39)	**1.2 (1.11–1.28) ***
Chromosomal/Congenital abnormalities				
No	ref	ref	ref	ref
Yes	**15.52 (8.91–27.02) ***	**7.9 (4.14–15.04) ***	**3.9 (1.97–7.7) ***	1.26 (0.92–1.72)
Smoking				
No	ref			
Yes	3.01 (0.99–9.13)			
Preterm history				
No	ref	ref	ref	ref
Yes	**7.23 (4.44–11.77) ***	**4.7 (3.04–7.28) ***	**3.5 (2.48–4.94) ***	**1.95 (1.68–2.26) ***
Pregnancy mode				
Spontanious pregnancy	ref	ref	ref	ref
Assisted pregnancy	**2.49 (1.19–5.21) ‡**	**2.82 (1.38–5.79) †**	**3.11 (1.9–5.08) ***	**1.61 (1.28–2.02) ***

Abbreviations: cOR, crude odds ratio; aOR, adjusted odds ratio; CI, confidence interval; Ref, referent; NA, not applicable; PTB, preterm birth; ETB, early term birth; DM, diabetes mellitus; GDM, gestational diabetes mellitus; PGDM, pre-gestational diabetes mellitus; PP, pre- and early-pregnancy; AGA, appropriate for gestational age; SGA, small for gestational age; LGA, large for gestational age; Apgar, appearance, pulse, grimace, activity, and respiration; NICU, neonatal intensive care unit; LR, labor room; OT, operation theatre. **Bold** values denote statistical significance at the * *p* < 0.001, † *p* < 0.01 and ‡ *p* < 0.05. ^a^ adjusted for maternal age, parity, nationality, diabetes status, chronic hypertension, baby gender, chromosomal/congenital abnormalities, smoking, preterm history, and pregnancy mode. ^b^ adjusted for maternal age, nationality, diabetes status, chronic hypertension, baby gender, chromosomal/congenital abnormalities, preterm history, and pregnancy mode. ^c^ adjusted for maternal age, nationality, education status, diabetes status, chronic hypertension, baby gender, chromosomal/congenital abnormalities, preterm history, and pregnancy mode. ^d^ adjusted for maternal age, parity, nationality, diabetes status, chronic hypertension, baby gender, chromosomal/congenital abnormalities, preterm history, and pregnancy mode. Notes: Only variables which showed significant results in the univariate analysis were included in the multiple logistic regression analysis, and presented in this table. PP obesity was exceptionally excluded from the multiple logistic regression, although significant in univariate analysis, due to missing values.

**Table 3 ijerph-18-05865-t003:** Multivariate regression analysis of the pregnancy and feto-maternal outcomes associated with PTB and ETB.

	LBW	SGA	LGA	Caesarean Delivery	Apgar < 7 at 1 Min	Apgar < 7 at 5 Min	In-Hospital Mortality	NICU/Death in LR/OT
	aOR (95%CI)	aOR (95%CI)	aOR (95%CI)	aOR (95%CI)	aOR (95%CI)	aOR (95%CI)	aOR (95%CI)	aOR (95%CI)
Extreme to very PTB ^a^	**5669.36 (1317.88–24,388.98) ***	**4.11 (2.55–6.64) ***	**3.8 (2.5–5.79) ***	**5.26 (3.71–7.47) ***	**52.3 (33.89–80.71) ***	**69.2 (20.54–233.09) ***	**84.73 (28.87–248.69) ***	**2417.24 (332.79–17,557.67) ***
Moderate PTB ^b^	**1021.64 (520.79–2004.19) ***	**3.78 (2.38–6.00) ***	**2.44 (1.63–3.64) ***	**4.54 (3.26–6.34) ***	**7.73 (4.43–13.50) ***	**9.81 (2.01–47.91) †**	**27.76 (7.52–102.48) ***	**274.38 (132.59–567.78) ***
Late PTB ^c^	**57.41 (40.1–82.19) ***	**1.9 (1.28–2.82) ***	**1.87 (1.39–2.5) ***	**2.47 (1.96–3.12) ***	1.64 (0.73–3.69)	NA	NA	**8.48 (6.37–11.28) ***
ETB ^d^	**5.96 (4.83–7.35) ***	0.9 (0.77–1.07)	**1.37 (1.24–1.52) ***	**2.14 (1.98–2.32) ***	**0.71 (0.51–0.99) ‡**	1.2 (0.45–3.19)	2.11 (0.84–5.32)	**1.31 (1.15–1.51) ***

Abbreviations: cOR, crude odds ratio; aOR, adjusted odds ratio; CI, confidence interval; Ref, referent; NA, not applicable; PTB, preterm birth; ETB, early term birth; DM, diabetes mellitus; GDM, gestational diabetes mellitus; PGDM, pre-gestational diabetes mellitus; AGA, appropriate for gestational age; SGA, small for gestational age; LGA, large for gestational age; Apgar, Appearance, Pulse, Grimace, Activity, and Respiration; NICU, neonatal intensive care unit; LR, labor room; OT, operation theatre. **Bold** values denote statistical significance at the * *p* <0.001, † *p* < 0.01 and ‡ *p* <0.05. ^a^ adjusted for maternal age, parity, nationality, diabetes status, chronic hypertension, baby gender, chromosomal/congenital abnormalities, smoking, preterm history, and pregnancy mode. ^b^ adjusted for maternal age, nationality, diabetes status, chronic hypertension, baby gender, chromosomal/congenital abnormalities, preterm history, and pregnancy mode. ^c^ adjusted for maternal age, nationality, education status, diabetes status, chronic hypertension, baby gender, chromosomal/congenital abnormalities, preterm history, and pregnancy mode. ^d^ adjusted for maternal age, parity, nationality, diabetes status, chronic hypertension, baby gender, chromosomal/congenital abnormalities, preterm history, and pregnancy mode.

## Data Availability

This is a research article and all data generated or analyzed during this study are included in this published article [and its Appendix A]. All enquiries should be directed to Nader Al-Dewik: naldewik@hamad.qa.

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
