# Peer review of "Incidence, Risk Factors, and Outcomes of Preterm and Early Term Births: A Population-Based Register Study"

_ijerph, 2021, doi:10.3390/ijerph18115865_

Round 1
Reviewer 1 Report
This is a very interesting study with information that may be useful in determinations of care of these babies.
Introduction:
line 48 - missing a reference
Methods:
Line 105: I am not sure what "immediate birth status" indicates, please explain further or in a different fashion.
Line 111: Please list maternal age in order, it is a bit confusing to go from 20-34 to <20 and then back to 25-39.
Line 118: I am not sure from where and what analysis you exclused women with history of multiples.
Results:
difficult to read overall and to follow
Tables in the result section are very difficult to read and need to be restructured to allow easier reading. Consider only using 2-3 tables with the most vital information and putting the rest in the supplemental data.
Discussion:
Line 251 - mortality of extreme and very PTB of 11% seems very low, considering you included down to 24 weeks GA. Do you mean 11% of all births, or of the preterm ones? Please explain better and consider including breakdown by gestation age.
Author Response
We would like to take this opportunity to thank the editor and the reviewers for their comments and for taking the time to evaluate our manuscript. We have revised the article accordingly and point-by-point responses to the comments are reported below. All the changes made to the manuscript are highlighted in the Main Document using the “Track Changes” tool.
Reply to each point by Reviewer 1
This is a very interesting study with information that may be useful in determinations of care of these babies.
line 48 - missing a reference
Response: references added for the paragraph: “Many of PTB born children who survive face adverse short-term and long-term consequences, including neurological impairments, chronic conditions, and in some cases, can lead to death” (Line 50).
Methods:
Line 105: I am not sure what "immediate birth status" indicates, please explain further or in a different fashion.
Response: sentence has been restructured. The phrase “immediate birth status” was deleted as the sentence meant to refer to Apgar score (Lines 106-107).
Line 111: Please list maternal age in order, it is a bit confusing to go from 20-34 to <20 and then back to 25-39.
Response: sentence has been restructured, as requested (Line 112-113).
Line 118: I am not sure from where and what analysis you excluded women with history of multiples.
Response: multiple births were excluded from “PTB history” variable to reduce selection bias, i.e., this variable does not include mothers who have multiple pregnancies. This was clarified in Line 119-121.
Results:
difficult to read overall and to follow
Tables in the result section are very difficult to read and need to be restructured to allow easier reading. Consider only using 2-3 tables with the most vital information and putting the rest in the supplemental data.
Response: tables were reconstructed as requested: univariate data from tables 2 and 3 were moved to supplementary materials Table S2 and Table S3, respectively, and only the multivariate analysis results were kept in the main tables. The results section was also rewritten at some parts (please see results section and tables).
Discussion:
Line 251 - mortality of extreme and very PTB of 11% seems very low, considering you included down to 24 weeks GA. Do you mean 11% of all births, or of the preterm ones? Please explain better and consider including breakdown by gestation age.
Response: 11.2% is the percentage of those born extreme to very preterm who died in the hospital (24/214, 11.2%) the remaining of those born extreme to very preterm (190/214, 88.8%) were discharged alive. We have clarified this in the discussion subsection and included mortality rates of the other PTB groups (Line 682-683).
Reviewer 2 Report
Incidence, Risk factors, and Outcomes of Preterm and Early Term Births: A Population-Based Register Study
This Paper used a large data set from the PEARL- study to look at outcome of PTB and ETB. The major contribution of this paper is the examination of the ETB population, as this is a relatively newly defined population. While the data and results are not unexpected, the report of the newly described ETB population is noteworthy.
Minor points
Is the any data on maternal drug and alcohol use during pregnancy? If so it may be worth adding this data.
In the tables a lot of numbers are reported as 0.000 as this number is bolded, I am taking that to read p <0.001. Please consider reporting this number as <0.001
Line 165 this is a missing comma after advanced maternal age( aOR.
Please define that other Nationalities are included in this study. While I assume this is a very diverse it would be beneficial to report some general population description. Possibly as simple as continental origin as a percent or the percent of self-reported race.
Author Response
We would like to take this opportunity to thank the editor and the reviewers for their comments and for taking the time to evaluate our manuscript. We have revised the article accordingly and point-by-point responses to the comments are reported below. All the changes made to the manuscript are highlighted in the Main Document using the “Track Changes” tool.
Reply to Reviewer 2 points:
Incidence, Risk factors, and Outcomes of Preterm and Early Term Births: A Population-Based Register Study
This Paper used a large data set from the PEARL- study to look at outcome of PTB and ETB. The major contribution of this paper is the examination of the ETB population, as this is a relatively newly defined population. While the data and results are not unexpected, the report of the newly described ETB population is noteworthy.
Minor points
Is the any data on maternal drug and alcohol use during pregnancy? If so it may be worth adding this data.
Response: Unfortunately, information on maternal drug and alcohol use during pregnancy is not available.
In the tables a lot of numbers are reported as 0.000 as this number is bolded, I am taking that to read p <0.001. Please consider reporting this number as <0.001.
Response: We changed the p values to “<0.001”, “<0.01” and “p<0.05” accordingly (please see relevant tables).
Line 165 this is a missing comma after advanced maternal age(aOR).
Response: A comma has been added (Line 482).
Please define that other Nationalities are included in this study. While I assume this is a very diverse it would be beneficial to report some general population description. Possibly as simple as continental origin as a percent or the percent of self-reported race.
Response: The distribution of other nationalities has been reported in the methodology section with numbers and percentages (please see Lines 115-117 page 3).
Reviewer 3 Report
Per the authors, fills a void in known data specific to Qatar (I cannot personally speak to this). Covers all expected risk factors for preterm birth across the GA spectrum.
However, this article reads really roughly. It's literally just a data dump, and predominantly tables. The results section should find a way to reference the table, and not simply repeat all the data points that are in the table. The discussion section was the most useful and understandable breakdown of the data, and the most readable section of the paper. Encourage the authors to find a way to condense their tables, and reflect on whether all of the data points need to be included, or if there are parts that can be streamlined for the main paper, and move more data to the supplemental tables. The majority of the pages of the paper should not be made up by tables.
Would also recommend in the first figure to add in the percentages instead of just the raw numbers.
You noted in the discussion that some of your data points had low absolute numbers for available data (ie employed status, consanguinity, etc). If the numbers are very low, this data is not useful to determine any conclusions, and may be some of the data that might be excluded from the tables.
Author Response
We would like to take this opportunity to thank the editor and the reviewers for their comments and for taking the time to evaluate our manuscript. We have revised the article accordingly and point-by-point responses to the comments are reported below. All the changes made to the manuscript are highlighted in the Main Document using the “Track Changes” tool.
Reply to Reviewer 3 points:
Per the authors, fills a void in known data specific to Qatar (I cannot personally speak to this). Covers all expected risk factors for preterm birth across the GA spectrum.
However, this article reads really roughly. It's literally just a data dump, and predominantly tables. The results section should find a way to reference the table, and not simply repeat all the data points that are in the table. The discussion section was the most useful and understandable breakdown of the data, and the most readable section of the paper. Encourage the authors to find a way to condense their tables, and reflect on whether all of the data points need to be included, or if there are parts that can be streamlined for the main paper, and move more data to the supplemental tables. The majority of the pages of the paper should not be made up by tables.
Response: Tables were reconstructed as requested, univariate data from tables 2 and 3 were moved to supplementary materials (Table S2 and Table S3 respectively), and only the multivariate analysis results were kept in the main tables. We have rewritten some parts of the results section to make it more clear.
Would also recommend in the first figure to add in the percentages instead of just the raw numbers.
Response: Done. We have added the percentages to Figure 1.
You noted in the discussion that some of your data points had low absolute numbers for available data (ie employed status, consanguinity, etc). If the numbers are very low, this data is not useful to determine any conclusions, and may be some of the data that might be excluded from the tables.
Response: The variables which had low data points were obesity, employment status, and consanguinity. These variables were excluded because they were unsuitable for the multiple regression model. They were also moved from the main tables to the supplementary materials (Table S2 and Table S3) to show their univariate relationships only and were not included in the multivariate analysis.
Round 2
Reviewer 1 Report
Thank you for addressing my concerns and adjusting the manuscript.